# Dot Symbol Auto-Filling Method for Complex Areas Considering Shape Features

**Yong Yin [1], Chengming Li [1,2,*] and Pengda Wu [2]**

[1] College of Geomatics, Shandong University of Science and Technology, Qingdao 266590, China; yinyong@casm.ac.cn

[2] Chinese Academy of Surveying and Mapping, Beijing 100830, China; wupd@casm.ac.cn

* Correspondence: cmli@casm.ac.cn; Tel.: +86-135-0119-0785

**Abstract:** Filling dot maps with a regular pattern is a key step in visual representation of land use data. The traditional methods cannot adapt well to shape features of complex areas, leading to an unreasonable symbol arrangement in the inner region and area boundaries during symbol filling. For this reason, a dot symbol auto-filling method of complex areas considering shape features is proposed in this paper. First, based on the constrained Delaunay triangulation, the internal structure of a complex area is divided into three simple filling areas denoted tile type, narrow type, and point type. Next, according to the geometric shape features, these three type areas are filled with plane, line, and point level of symbols respectively. Finally, the dot symbols near to boundaries are adjusted on the basis of the boundary constraint to optimize the symbol-filling effect. Based on the national data of a region in Guizhou Province, the method proposed in this paper is compared to the traditional symbol filling methods for validation. The experimental results show that the proposed method improves the dot symbol sufficiency of complex areas, and the edge of dot symbol group closely adhere to the boundary and conform to the extension characteristics of boundary without creating spatial conflicts such as spatial overlap, the filling result can better reflect the shape features of the areas.

**Keywords:** complex areas; dot symbol auto-filling; Delaunay triangulation; fine division; boundary constraint

## 1. Introduction

Dot symbol filling is a key step in visual representation of land use data [1,2]. Generally, its objective is to set a certain interval distance in closed polygonal areas to continuously draw a specific pattern to convert the map elements into user-recognizable pictograms and describe the map content of the filled area intuitively and vividly [3,4]. In the process of map symbol filling, it is necessary to properly configure the map symbols, considering the features of the filled area to adaptively characterize the spatial pattern distribution and the variation law of geographic elements [5–8].

Lavin [9] developed the "dot-density shading" method using regular grid to finish random dot placement with limited dot overlap. However, it focuses on continuous data and it cannot be applied to discrete data without adaptations. Kimerling [10] published another method of random dot placement. He uses the Mackay [11] nomograph and correlates dot density and dot size. However, this method leads to a reduced legibility of the map because of the overlap of dots interferes. Harrie and Revell [12] developed a method for optimizing the semi-random placement patterns to achieve the automation of vegetation symbol placement for Ordnance Survey 1:50,000 scale maps. Hey [13] proposed a method of placing dots in a centered manner as dot clusters grouping

around one central point, this method connects the area needed for placing the dots with the number and size of the dots. The above method makes dot maps with non-regular pattern to show details in the spatial variation of geographic phenomena such as human populations or agricultural crops. For the land use thematic elements, such as cultivated land, forested land, and garden land, each element area is considered to be the smallest and indivisible geographic unit within homogeneous land use class. The symbols are placed inside areas to represent polygon features, so the dot map with regular pattern placement is preferable in the visual representation of land use data, such as "three-square type" or "pound type" pattern [14,15]. Kimerling [10] solved the graphic spatial overlap problems at the boundary of a surface due to the increase in the load of map elements in the symbol filling process. Since the "three-square shaped" or "pound shaped" symbols are laid out according to the extended regular texture of the surface, they can handle the regular areas in the plain area well [16]. However, the shape of land use areas is greatly influenced by the characteristics of the natural geographical environment. The terrains of low hills and valleys are mostly complex areas with various shapes, widths, narrowness, and undulations. Jiang et al. [17] pointed out that if the above-mentioned areas were filled according to the "three-square shaped" or "pound shaped" symbols, there would be problems such as uneven symbol layout in the planar area and few or no symbols in the narrow area. For this reason, a symbol-filling method for complex areas has been proposed by Jiang et al. [17] based on simple polygonal segmentation. This method relies on the concept of a simplex to divide the land-type areas into simplexes based on the constrained Delaunay triangulation. Accordingly, the area is divided into two parts, a narrow part and a two-dimensional extension part, that are filled by type. However, this method still cannot adapt well to the shape features of complex areas, and the segmentation of complex areas is somewhat rough, leading to the problem that the two-dimensional extension also shows fewer or no symbols in some regions. Additionally, the filled symbols at the boundary will not lead to conflicts, such as spatial overlap, but the generalization of the boundaries is inaccurate. Based on the above analysis and the existing research, a dot symbol auto-filling method of complex areas is proposed. The aim of this paper is to enhance the sufficiency and rationality of dot symbol filling of complex areas through the fine division of areas and symbol correction under boundary constraint, such that the filling results can reflect the shape features of these areas better.

The remainder of this paper is organized as follows. Section 2 reviews the related works and analyzes the advantages and disadvantages of existing dot symbols filling method. Section 3 describes the methodology of a dot symbol auto-filling method for complex areas considering shape features. Section 4 demonstrates the experiments and analyses. Section 5 presents the conclusions.

## 2. Related Works

### 2.1. Existing Dot Symbols Filling Method

Jiang et al. [17] presented a dot symbol filling method for the complex land-type areas based on a simple polygonal segmentation. The basic idea of the method is constructing constrained Delaunay triangulation [18] for complex areas, and a simplex segmentation is performed, such that the narrow area and the two-dimensional extension area of a polygon can be distinguished according to the geometric features of the polygon. Then, various strategies are designed to fill the dot symbols according to the characteristics of different parts of the polygon. The specific steps are as follows.

Step 1: Based on the constrained Delaunay triangulation, a simple segmentation is performed on the complex land-type area.

Step 2: The triangles in the triangulation are divided into three categories, and the skeleton lines of polygons are extracted from each triangle.

Step 3: Calculating the width of each type of triangle to obtain the average width of a simplex sequence corresponding to each segment of the skeleton line.

Step 4: Setting the width threshold, the simplex sequence whose width is greater than the threshold is defined as the two-dimensional extension area and the other simplex sequence is defined as the narrow area.

Step 5: For the two-dimensional extension area, the filling is performed based on the traditional three-square type and the spatial overlap problem of symbols and boundaries is solved using the method proposed by Nie et al. [19].

Step 6: For the narrow area, the filling interval of the dot symbol is determined first, and then the central axis with a single line interval is used for filling [20].

### 2.1.1. Three-Square Type Symbol

The basic definition of three-square type symbol is as follows:

(1) If the width of the three-square symbol is denoted by $w$, then the footprint ($a$) of one symbol is $a = w \times w$.

(2) The top-to-bottom and left-to-right intervals between the three-square type symbols are denoted as $q$, shown by the blue line in Figure 1.

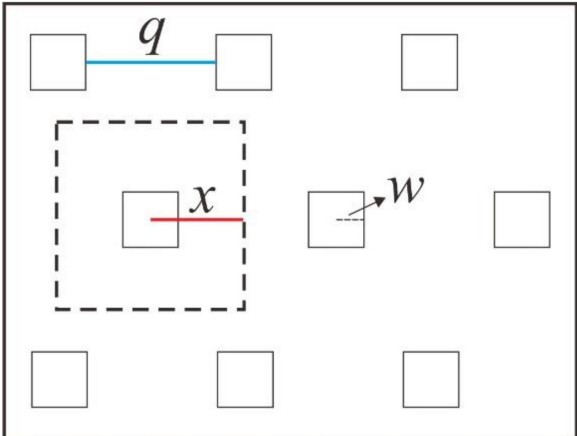

**Figure 1.** Schematics of three-square shaped filling.

(3) The horizontal offset of the three-square type symbol is denoted as $x$; then, $x = \frac{1}{2}(w + q)$, as shown by the red line in Figure 1.

(4) The influence domain of the symbol [21] is denoted by $s$ and is computed as $s = (2x)^2 = (w + q)^2$, as shown by the dotted rectangle in Figure 1.

### 2.1.2. Average Width of a Triangle Set

A complex area is segmented based on the Delaunay triangulation. After the skeleton line has been extracted and corrected, the average width $\overline{w}$ of each triangle set can be calculated as follows [22]:

$$\overline{W} = \sum_{i=1}^{k} \frac{|P_i P_{i+1}|}{L} |W_{i1} W_{i2}| \tag{1}$$

where $L$ is the total length of the skeleton line corresponding to the triangle set, $k$ is the number of triangles that run through the skeleton line, $|P_i P_{i+1}|$ is the length of the skeleton line that is within the triangle, and $|W_{i1} W_{i2}|$ is the width of the triangular element.

The triangles in the Delaunay triangulation can be divided into three types according to the number of adjacent triangles [23,24]:

- Type-I triangle: There is only one adjacent triangle, and the two sides of the triangle are the boundaries of the polygon.

- Type-II triangle: There are two adjacent triangles, which is the backbone structure of the skeleton line and describes the extension direction of the skeleton line.
- Type-III triangle: There are three adjacent triangles, which are the intersections of the skeleton line branches as the starting points for stretching in three directions.

As shown in Figure 2, the width of a triangular element is defined as follows: the width of the Type-I triangle is the distance between the midpoints of the two Type-I sides, as shown by $|W_{k1}W_{k2}|$ in Figure 2; the width of the Type-II triangle is the distance from the midpoint of the Type-I side to the corresponding vertex, as shown by $|W_{i1}W_{i2}|$ in Figure 2; and the width of the Type-III triangle is the length of the intersected side of the skeleton line, as shown by $|W_{l1}W_{l2}|$ in Figure 2.

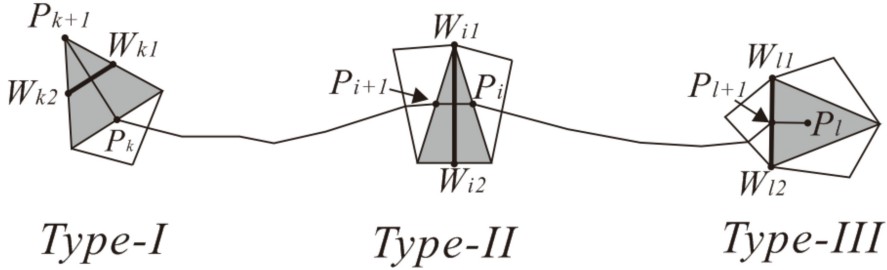

**Figure 2.** Triangular element width.

*2.2. Deficiencies of the Existing Method*

The existing dot symbols filling method for the complex areas is mainly based on two parameters: (1) the segmentation of the skeleton line by the position of the Type-III triangle, and (2) distinguishing each segment of the skeleton lines by the width threshold. However, this method is inaccurate for segmentation of complex areas. This is specifically shown as follows:

(1) Consider the region that visually appears as a long and narrow area, it will be divided into multiple scattered regions due to a large variation of wide and narrow parts.

(2) It is impossible to distinguish the branches without the participation of Type-III triangles.

(3) The width threshold is difficult to determine, and if the threshold is too large or too small, the segmentation of the narrow portion and the two-dimensional extension portion may not be sufficiently thorough.

In addition, although the existing research considers the spatial overlap conflict at the boundary, the filled symbols cannot adequately summarize the boundaries, and a gap exists in the visual cognition requirements.

## 3. Methodology

In this paper, an auto-filling method of complex areas considering shape features is proposed, including three key steps: (1) fine division of the internal structure: the internal structure of a complex area is divided into a plurality of simple filling areas based on constrained Delaunay triangulation; (2) symbol filling: according to the geometric features of the segmentation, it is divided into three types—tile type, narrow type, and point type—for symbol filling at the point, line, and plane levels, respectively; and (3) symbol correction under boundary constraint: the symbol at the boundary of the area is adjusted based on the inner and outer buffer constraint algorithm to optimize the boundary symbol filling effect. Our method has four input parameters, the width of the dot symbol ($w$), the dot symbol space ($q$), and the width and area thresholds ($W_T$ and $A_T$).

### 3.1. Fine Division of Internal Structure

### 3.1.1. Branch Extraction

Step 1: A constrained Delaunay triangulation is created for the complex land-use area, where the initial skeleton line is extracted based on the triangle classification, as shown in the blue solid line in Figure 3a, and the topology is constructed according to the initial skeleton line.

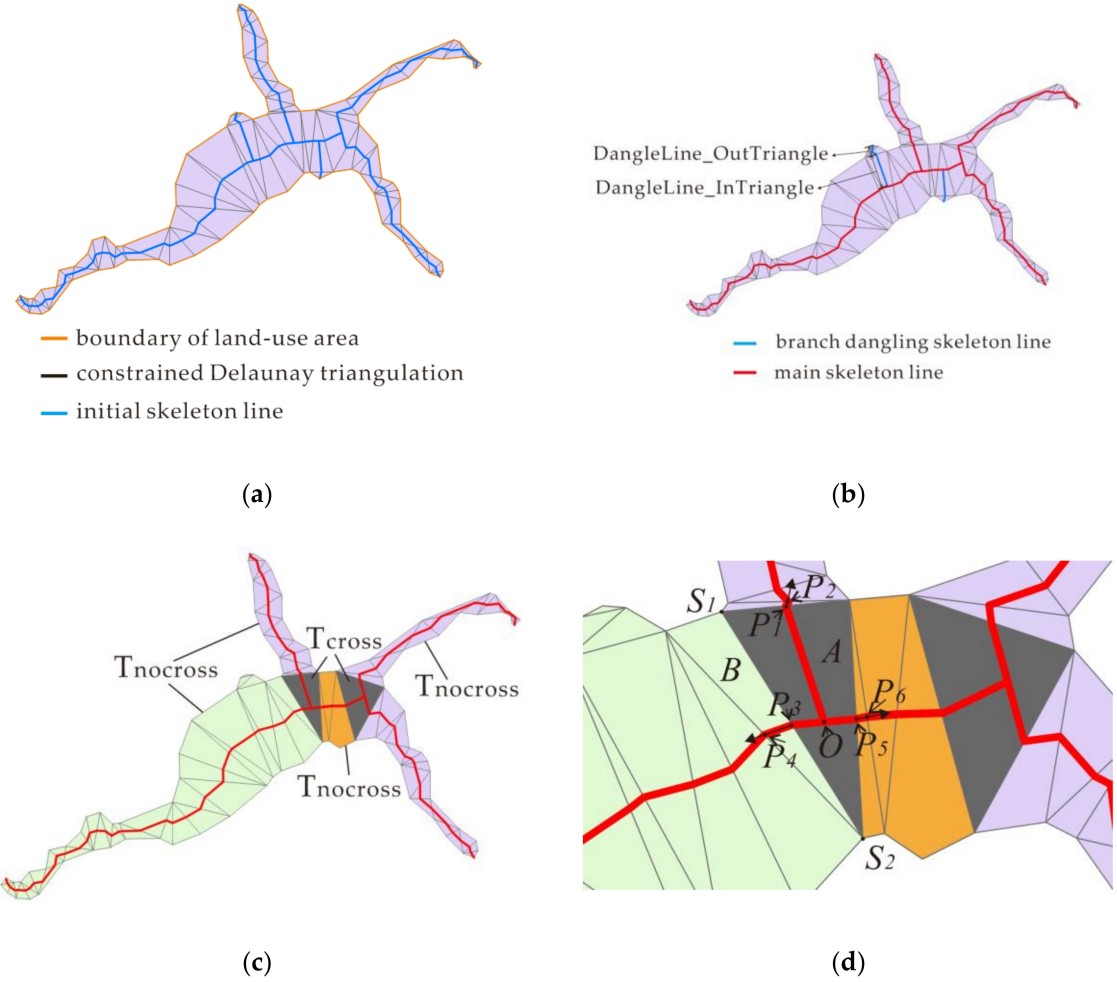

**Figure 3.** Branch extraction for a complex area. (**a**) Triangle network and skeleton line of a complex area. (**b**) Eliminating the branch dangling lines. (**c**) Identifying the branch triangles. (**d**) Determining the assignment of Type-III triangles.

Step 2: The branch dangling skeleton line is pruned according to the minimum symbol fill limit. The branch dangling skeleton line consists of two parts. The inner section of the Type-III triangle is denoted by DangleLine_InTriangle, the length of which is denoted by $L_{in}$; the outer section of the Type-III triangle is denoted by DangleLine_OutTriangle, the length of which is denoted by $L_{out}$, and the area is denoted by $S_{out}$. The branch pruning first satisfies $L_{out} < L_{in}$ followed by $L_{out} < 2x$ or $S_{out} < s$, as shown by the blue solid line in Figure 3b.

Step 3: Based on the intersection relationship of skeleton lines (the number of skeleton lines to the node being greater than 2), the Type-III triangle $T_{cross}$ where the intersection is located is first identified, and the Type-III triangles set {$T_{cross(i)}$, i = 1, 2, . . . , n} is obtained. Additionally, the branch triangles set {$T_{nocross(i)}$, i = 1, 2, . . . , n} is obtained by difference operation between {$T_{cross(i)}$, i = 1, 2, . . . , n} and the whole triangles in the area, as shown in Figure 3c.

Step 4: The Type-III triangle assignment is determined. For any Type-III triangle in set $\{T_{cross}\}$, the algorithm for determining which $T_{nocross}$ can be connected with it is as follows. For each Type-III triangle, obtaining three nodes on each side and connect each node with its adjacent node on the branch skeleton line to form three vectors, the angles for any two of the vectors are calculated. The two vectors corresponding to the largest angles are considered as the extension direction of the skeleton line. The longest side of $T_{cross}$ corresponding to the two vectors is taken; $T_{cross}$ belongs to $T_{nocross}$ with the co-edge of the longest side. As shown in Figure 3d, for the Type-III triangle A, obtaining three vectors $\overrightarrow{P_1P_2}$, $\overrightarrow{P_3P_4}$, $\overrightarrow{P_5P_6}$, and the angle between $\overrightarrow{P_3P_4}$ and $\overrightarrow{P_5P_6}$ is the largest angle, therefore it is considered the extension direction. $S_1S_2$ is the longer side corresponding to the two vectors, so the Type-III triangle A belongs to adjacent triangle B.

### 3.1.2. Segment Extraction

Segment extraction refers to dividing the branch of the complex area into several segments, and the triangles within each segment have a similar width. The specific steps are as follows.

Step 1: The extracted branch is taken as a basic element. The above-mentioned calculation method of the triangle element width is used to calculate the width of each triangle in the branch. Setting width threshold, a triangle whose width is greater than the width threshold is defined as a wide triangle and a triangle whose width is less than or equal to the width threshold is defined as a narrow triangle. In this paper, the width threshold is given by *2x*, where *x* is the horizontal offset of the three-square type symbol.

Step 2: The initial clustering is performed according to the triangle being wide or narrow; the adjacent narrow triangle lines are placed into the same narrow triangle set $NS_i$, and the adjacent wide triangles are put into the same wide triangle set $WS_i$.

Step 3: The lengths of the skeleton line in all the triangle sets in the branch are counted. The length of the skeleton line in the narrow triangle line set is denoted as *L1*, and that in the wide triangle line set is denoted as *L2*; their ratio *L1/L2* is calculated. If the ratio $\geq 1$, clustering is performed on each triangle set as follows to achieve further segmentation of the branch triangle:

(1) A narrow triangle set is aggregated into the adjacent wide triangle set. Whether both sides of the narrow triangle set are wide triangle sets is determined. If they are all wide triangle sets, it can be determined whether the length of the skeleton line in the narrow triangle set is smaller than the length of the skeleton line in the wide set on the left and right sides. If so, the narrow triangle sets are merged into the wide triangle set on the left and right sides; otherwise, no merging is performed.

(2) Wide triangle sets are aggregated into adjacent narrow triangle sets. Whether both sides of the wide triangle set are narrow triangle sets is determined. If they are all narrow triangle sets, it is determined whether the length of the skeleton line in the wide triangle set is smaller than the length of the skeleton line in the narrow set on the left and right sides. If so, wide triangle sets are merged into the narrow triangle set on the left and right sides directly; otherwise, no merging is performed.

(3) End wide and narrow triangle sets are aggregated. If a triangle set only exists on one side adjacent to a certain triangle set, it is an end triangle set, and the lengths of the skeleton lines at the end wide and narrow triangles are calculated. If it is smaller than $w + q$, it is merged into the adjacent triangle set.

As shown in Figure 4a, the branch is divided into four segments, A, B, C, and D. After aggregation, they are merged into two segments, A and D, as shown in Figure 4b.

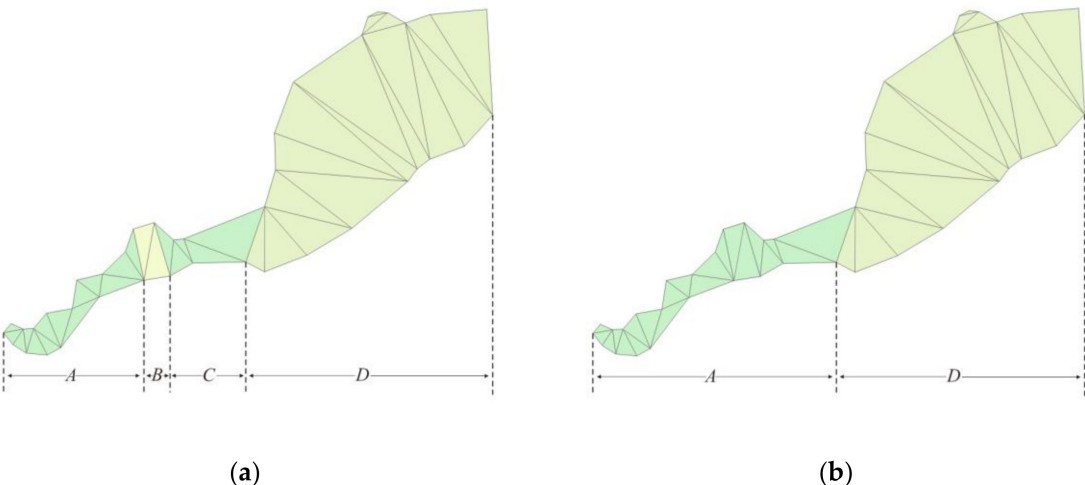

(**a**)             (**b**)

**Figure 4.** Segment extraction of a complex area. (**a**) Identification of segmented area. (**b**) Aggregation of segmented area.

### 3.2. Symbol Filling

The segments are divided into three types—tile type, narrow type, and point type—according to the geometric width and area features. Setting width and area thresholds, $W_T$ and $A_T$, and calculating the average width and area of each segment, $W_S$ and $A_S$, then each segment is classified according to the following rules.

- If $W_S \geq W_T$ and $A_S \geq A_T$, the segment belongs to tile type.
- If $W_S < W_T$ and $A_S \geq A_T$, the segment belongs to narrow type.
- If $W_S \geq W_T$ and $A_S < A_T$, the segment belongs to point type.
- If $W_S < W_T$ and $A_S < A_T$, the segment does not need to fill.

Generally, the initial value of $W_T$ is equal to twice width of the three-square symbol, i.e., $2w$, and the initial value of $A_T$ is equal to twice the influence domain area of the symbol, i.e., $2s$.

Considering the shape features, different rules are used to fill the dot symbol of various regions. The tile type is filled with the traditional method of three-square type symbol which is with uniform filling along the extended layout; the narrow type is filled with an equal spacing of points along the central axis to harmonize the overall filling such that the filling spacing is equal to the symbol spacing of the tile type. The point type is filled with a single symbol.

The specific idea is to obtain the origin point $(x_0, y_0)$ for symbol filling and calculate the minimum bounding rectangle (MBR) of the area. Generally, the origin point $(x_0, y_0)$ is the lower left point of MBR; for convenience, it can be set to $(0, 0)$. The starting point $(x_{start}, y_{start})$ and ending point $(x_{end}, y_{end})$ of the dot symbols are calculated via the following equations:

$$x_{start} = \text{int}(\frac{x_{min} - x_0}{\Delta x}), \ y_{start} = \text{int}(\frac{y_{min} - y_0}{\Delta y}) \tag{2}$$

$$x_{end} = \text{int}(\frac{x_{max} - x_0}{\Delta x} + 1) \tag{3}$$

where int(*) indicates the integral function; $\Delta x = q$ and $\Delta y = q$; $(x_{min}, \ _{smin})$ is the lower-left coordinate of MBR, and $(x_{max}, y_{max})$ is the upper-right coordinate of MBR.

The starting point $(x_{start}, y_{start})$ and the endpoint $(x_{end}, y_{end})$ of the filling are traversed, and the offset is applied to the upper and lower interval traversals. Additionally, the buffer of $w/2$ out of the area surface is used to determine whether the fill point is valid. It is noted that there will be redundant fill points in the range of $w/2$ outside the boundary for the three-square type filling of the area.

*3.3. Symbol Correction under Boundary Constraint*

Symbol filling at the boundary is a difficult problem in the visual representation of complex areas. On the one hand, the filled symbols at the boundary easily lead to conflicts, such as spatial overlap; on the other hand, the filled symbols cannot adhere to the boundary closely, leading to no or fewer symbols at the boundary. The former problem has been studied and solved [10], therefore this paper focus on the latter problem. The flow chart of symbol correction under a boundary constraint is shown in Figure 5.

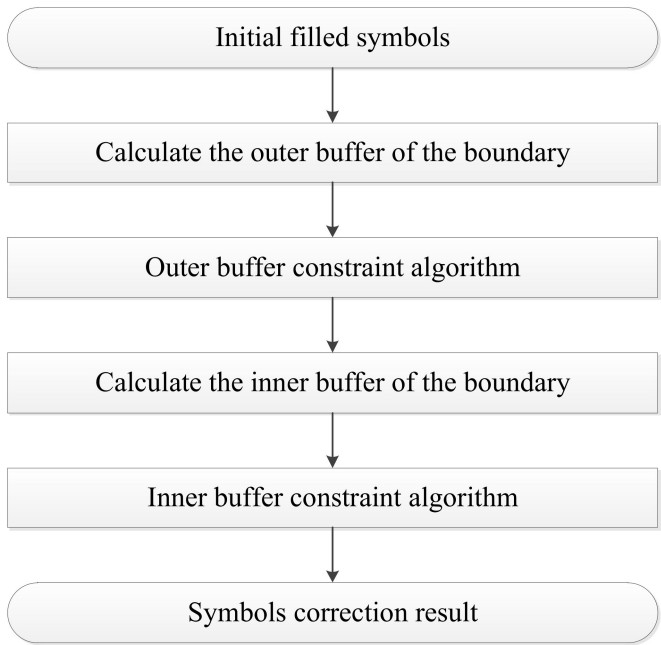

**Figure 5.** The flow chart of symbol correction under boundary constraint.

As depicted in Figure 5, symbol correction is implemented by calculating the boundary outer and inner buffers and their respective buffer constraint algorithms. Symbols located outside the area and adjacent to the boundary can be moved from outside to inside via the outer buffer constraint algorithm. Symbols located inside the area and adjacent to the boundary can keep an appropriate distance from the boundary by the inner buffer constraint algorithm.

3.3.1. Outer Buffer of Boundary Constraint

Step 1: Outer buffering is performed on the area's boundary ($B$) according to buffer = $w/2$ to obtain the outer buffer surface ($B_{out}$).

Step 2: Whether the three-square shaped fill point p of the area is in $B_{out}$ is determined; if there are points in $B_{out}$, the shift operation will be performed.

Step 3: The specific processing method of shifting is shown in Figure 6. First, the nearest neighbor point $p_o$ from $p_{out}$ to $B$ is calculated; next, the symmetry point $p_s$ of $p_{out}$ is calculated based on point $p_o$, and $p_s$ is the alternative point of $p_{out}$ under the constraint of the outer buffer of the boundary.

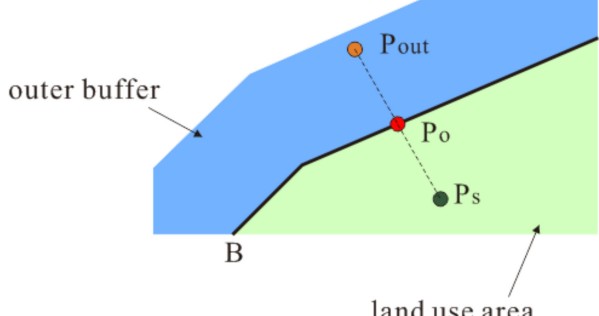

**Figure 6.** Schematic diagram of symbol point shift under the outer buffer constraint.

### 3.3.2. Inner Buffer of Boundary Constraint

Step 1: The inner buffer is constructed on the area boundary ($B$) according to buffer = $w/2$ to obtain the inner buffer surface ($B_{in}$).

Step 2: Whether the three-square shaped fill point p of the area is in $B_{in}$ is determined; if there are points in $B_{in}$, the shift operation will be performed.

Step 3: The specific processing method of shifting is shown in Figure 7. First, the nearest neighbor point $p_o$ from $p_{in}$ to $B$ is calculated; next, the direction of vector $\overrightarrow{p_o p_{in}}$ is taken as the shift direction. Point $p_{in}$ moves to $p_s$ along the shift direction by $w/2$ from the boundary, and $p_s$ is the alternative point of $p_{in}$ under the constraint of the inner buffer of the boundary.

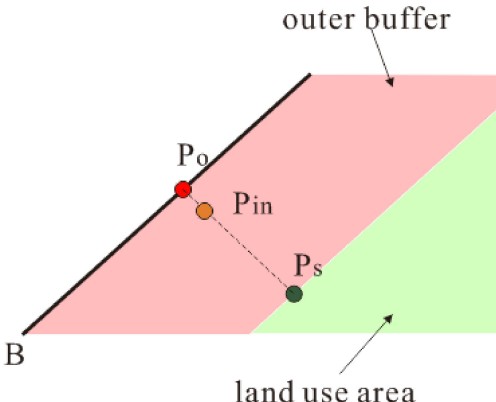

**Figure 7.** Schematic diagram of symbol point shift under the inner buffer constraint.

Figure 8 is an example of symbol filling of a certain area under boundary constraints. Figure 8a shows the initial tree-square shaped fill effect. It is observed that some symbol points are outside the boundary of the area. These points, resulting from the application of the outer buffer constraints, are shifted to the inside of the area, as shown in Figure 8b,c. After the inner buffer constraint has been applied, the point near the boundary can maintain a moderate distance from the boundary, as shown in Figure 8d,e; the final filling result is shown in Figure 8f.

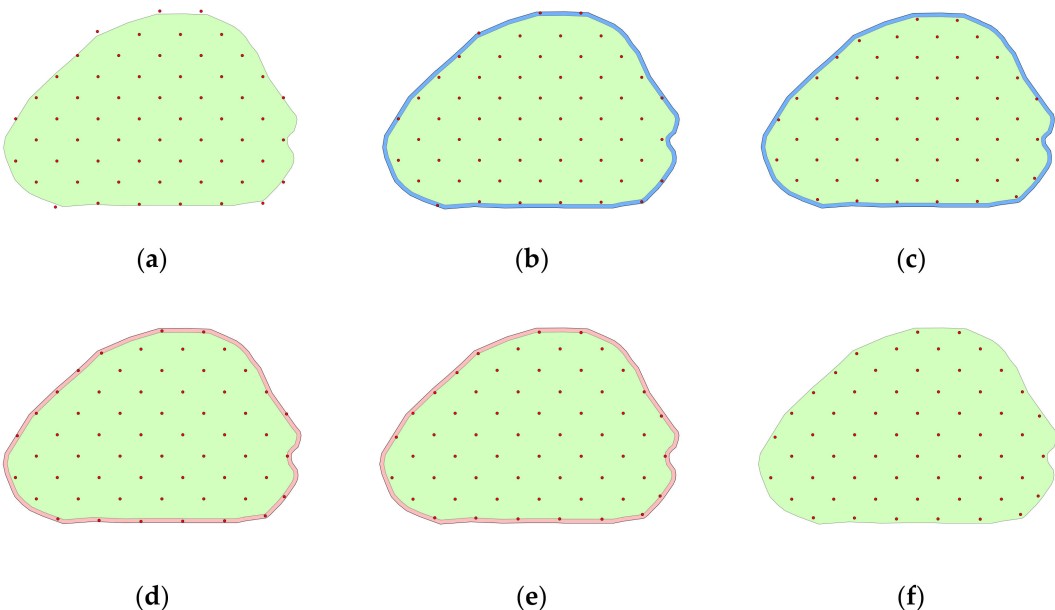

**Figure 8.** Filling under the constraint of boundary features. (**a**) Initial three-square shaped filling. (**b**) Outer buffer constraint. (**c**) Processing result of the outer buffer constraint. (**d**) Inner buffer constraint. (**e**) Processing result of the inner buffer constraint. (**f**) Filling result after the boundary constraint has been applied.

## 4. Experiment and Analysis

### 4.1. Experimental Data and Environment

The dot symbol auto-filling method of complex areas considering shape features proposed in this paper has been implemented using the WJ-III map workstation developed by the China Institute of Surveying and Mapping. WJ-III map workstation has a large number of map data production libraries such as geometric calculation, topological processing, etc. (http://www.casm.ac.cn/cgzh.php?col=142&file=4129). Our method was compared to the traditional symbol-filling method based on a simple polygonal segmentation to verify the validity and rationality. The experimental environment is a 64-bit Microsoft Windows 7 operating system with an Intel Core I7-3770 CPU operating at the main frequency of 3.2 GHz, 16 GB of RAM, and 1024 GB solid state hard disk.

### 4.2. Typical Data Analysis

#### 4.2.1. Rational Filling of Complex Areas

The polygon shown in Figure 9 is a complex area given in Jiang et al. [17]. The polygons shown in Figures 10 and 11 are two pieces of real-world land-type area data. The three polygons are all meandering with complex structures, where extended areas coexist with narrow branches. These three polygons were used to verify the reliability and rationality of proposed method for filling complex areas.

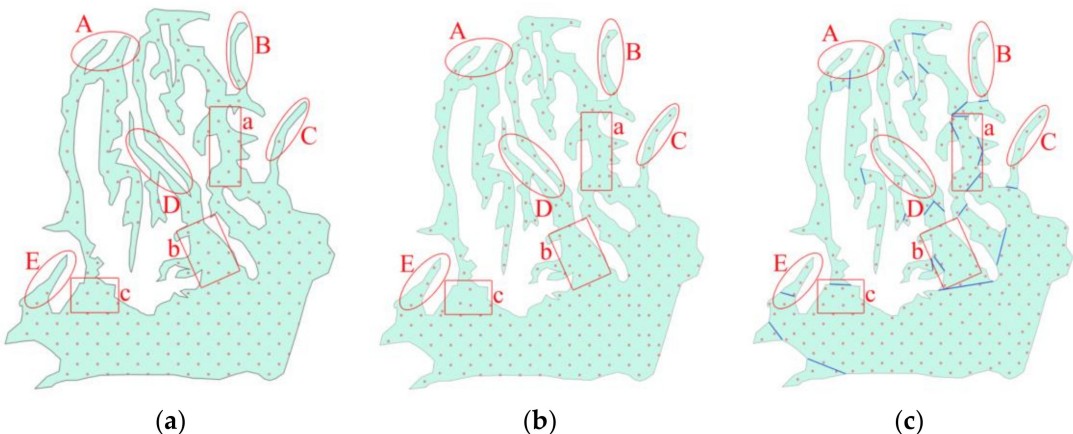

**Figure 9.** Complex area 1 (the blue solid line is the segmentation line). (**a**) Filling using the traditional three-square type method. (**b**) The effect of the filling method based on simple segmentation. (**c**) The effect of the method proposed in this paper.

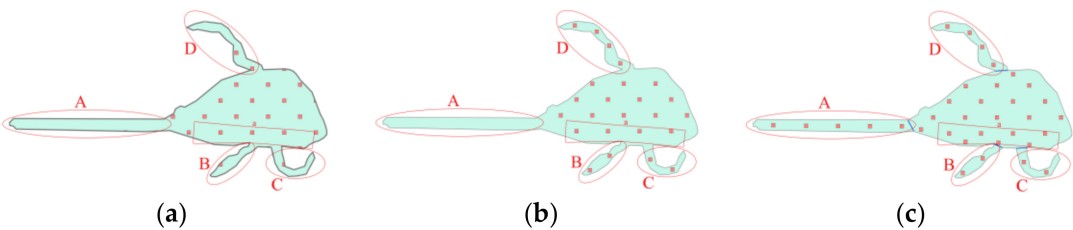

**Figure 10.** Complex area 2 (the blue solid line is the segmentation line). (**a**) Filling using the traditional three-square type method. (**b**) The effect of the filling method based on simple segmentation. (**c**) The effect of the method proposed in this paper.

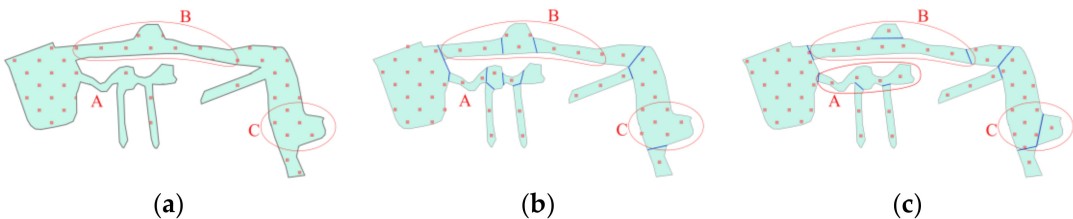

**Figure 11.** Complex area 3 (the blue solid line is the segmentation line). (**a**) Filling using the traditional three-square type method. (**b**) The effect of the filling method based on simple segmentation. (**c**) The effect of the method proposed in this paper.

Figure 9a shows that in the narrow parts of a complex area (elliptical areas A, B, C, D, and E), the traditional three-square type direct filling method resulted in fewer or no symbols. The filling method based on a simple segmentation and the method proposed in this paper could better identify these narrow regions such that the symbols were filled in these regions, and the shape of the symbol arrangement was basically consistent with the long and narrow area. In addition, the wide and narrow parts of the complex area were more apparent (rectangles a, b, and c). The method proposed in this paper resulted in a more uniform filling and a more reasonable overall layout than those of the filling method based on a simple segmentation. The filled symbols were more refined in the shape features of the region, which is consistent with the law of human visual cognition.

Figure 10 shows that although a certain number of branches can be processed by the filling method based on simple segmentation, as shown in ellipses B, C, and D in Figure 10b, the segmentation of the branches depended on the Type-III triangles among the Delaunay triangles. Therefore, the narrow part of the extended area could not be identified, as shown in ellipse A in Figure 10b. This method could identify such a branch and fill it, as shown in ellipse A in Figure 10c. In addition, at the boundary of

a complex area, the filling effect of the filling method based on a simple segmentation was too sparse, while that of the proposed method was more compact, as shown in rectangular box a in Figure 10.

Figure 11 shows that the result of the method of this paper was completely different from the segmentation result of the filling method based on simple segmentation. The latter only depended on the segmentation method of the width threshold that can segment complex areas. However, the segmentation was too fragmented, and the segmentation result could not maintain the extension characteristics of the original branch. In contrast, the segmentation result of the method proposed in this paper applied to the complex area better preserved the shape feature of the original area, and there were no cases of fewer or no symbols. The sufficiency of symbol filling of the area was also significantly higher than that of the other two methods, as shown in Ellipses A, B, and C in Figure 11.

### 4.2.2. Rational Filling of Regular Areas

The polygons shown in Figures 12–14 are three land use areas with regular shape. The main body of them belonged to the tile type and without branches. These three polygons were used to verify the reliability and rationality of the proposed method for filling regular areas.

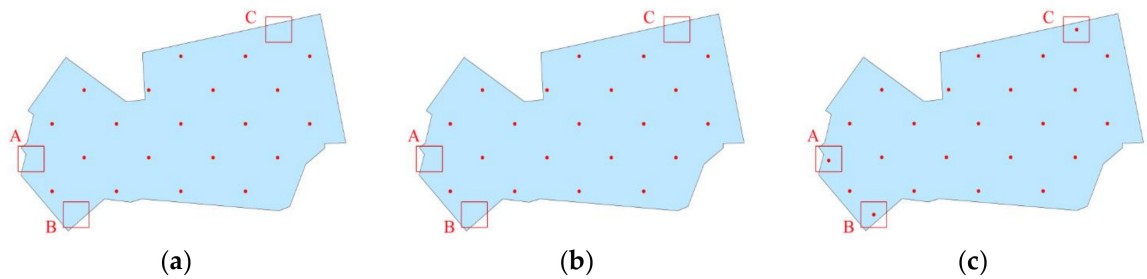

(**a**)            (**b**)            (**c**)

**Figure 12.** Regular area 1. (**a**) Filling using the traditional three-square type method. (**b**) The effect of the filling method based on simple segmentation. (**c**) The effect of the method proposed in this paper.

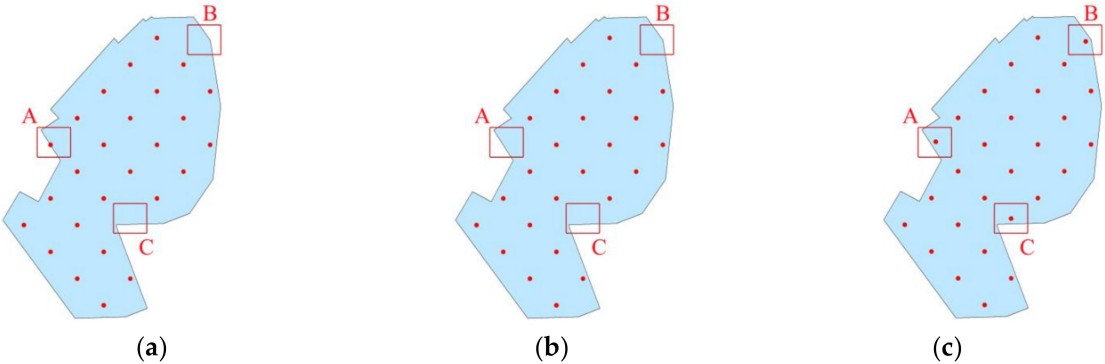

(**a**)            (**b**)            (**c**)

**Figure 13.** Regular area 2. (**a**) Filling using the traditional three-square type method. (**b**) The effect of the filling method based on simple segmentation. (**c**) The effect of the method proposed in this paper.

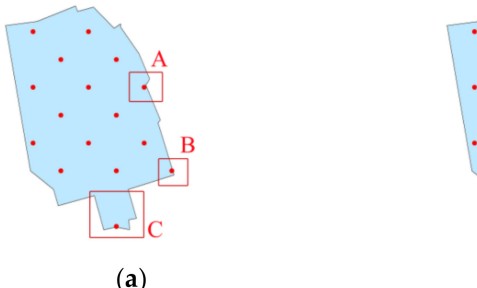 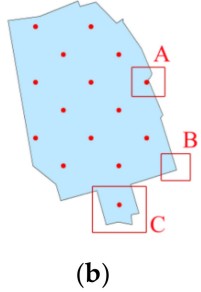 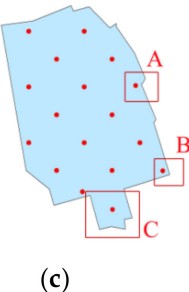

(**a**)  (**b**)  (**c**)

**Figure 14.** Regular area 3. (**a**) Filling using the traditional three-square type method. (**b**) The effect of the filling method based on simple segmentation. (**c**) The effect of the method proposed in this paper.

As shown in Figures 12–14, the filling effect of three method is basically the same because there is no branch structure interfering with filling. The dot symbols cover the whole region of the areas well. However, for the dot symbols at the boundary, the traditional three-square type method resulted in spatial overlap and an unreasonable dot distribution, the filling method based on simple segmentation solved the problem of spatial overlap but the symbols are still unreasonable, and the symbols filled by our method fit the boundary shape well and reflected the polygon shape more accurately.

*4.3. Mass Data Analysis*

Considering the forested land type in the census data of a city in Guizhou Province as an example, the reliability of the method applied to a large data volume was analyzed. The experimental data contained a total of 174 forest areas that were large in size and complex in shape. The symbol interval was set to 500 m with the symbol width of 50 m. The number of dot symbols, sufficiency, symbol overlap, cover, and the number of spatial conflicts using the conventional three-square shaped filling method, the filling method based on simple segmentation, and the method of this paper are shown in Table 1.

**Table 1.** Comparative analysis of the reliability of methods applied to a large data volume.

| Filling Method | Number of Dot Symbols | Sufficiency | Symbol Overlap | Number of Spatial Conflicts |
| --- | --- | --- | --- | --- |
| Traditional three-square shaped filling method | 14,032 | 0.67 | 0.64 | 1148 |
| The filling method based on simple segmentation | 17,650 | 0.88 | 0.78 | 745 |
| The method proposed in this paper | 20,034 | 0.95 | 1 | 0 |

In the table, the sufficiency of dot symbols is denoted by $F$ and was computed using $F = (s \times n)/S$. In the preceding equation, $s$ is the single-symbol influence domain, $S$ is the area of the filled region, and $n$ is the number of symbols in the filled region. Due to the particularity of a boundary symbol's position, its influence domain is considered to be $1/2s$. Symbol overlap refers to the ratio of the number of symbols obtained by the filling method to the number of symbols of the method proposed in this paper. The number of spatial conflicts refers to the number of symbols that are covered with the boundary during the filling.

Table 1 shows that the number of dot symbols in the experimental area was 20,034, which was the maximum for the same symbol interval and size. The sufficiency of the method proposed in this paper was 0.95, indicating that this method was more compact than other methods for area filling with a better reflection of the complex area shape. Among the other two methods, the symbol overlap of the traditional three-square shaped filling method was 0.64, indicating that there were more narrow areas to be filled; the symbol overlap of the filling method based on a simple segmentation was 0.78, indicating that some areas were still not effectively segmented. In addition, because the boundary constraint was considered by the method proposed in this paper, the symbols filled by this method

will not collide with the boundary, generating spatial conflicts such as cover. Therefore, the number of spatial conflicts was zero.

The computing time of the conventional three-square shaped filling method, the filling method based on simple segmentation, and the method of this paper are shown in Table 2.

**Table 2.** Comparative analysis of the computing time of methods applied to the large data volume.

| Filling Method | Dataset 1 | Dataset 2 | Dataset 3 |
|---|---|---|---|
| Number of polygons processed | 174 | 3050 | 5296 |
| Total area of polygons ($km^2$) | 432 | 1211 | 1632 |
| Total computing time (s) | 2.12 | 27.69 | 59.95 |

The computing time of the three experiment areas of our method was 2.12 s, 27.69 s, and 59.95 s, and the average computing time of a single area was about 0.01 s.

## 5. Conclusions

The traditional symbol filling methods cannot adapt well to the shape features of a complex area, leading to an unreasonable symbol arrangement in the inner region and boundary of the symbol when the dot symbols with a regular pattern are filled. For this reason, an auto-filling method of a complex area considering shape features is proposed in this paper, which not only ensures that the symbols inside the area are evenly distributed and compactly arranged, but also guarantees the rationality of the symbols at the boundary. A verification using an actual dataset results in the following conclusions:

(1) The filling result of complex and regular areas obtained using the method proposed in this paper can better reflect the shape feature and distribution structure of the original area, and the edge of the dot symbol group fit the boundary of the area better.

(2) The sufficiency of the symbols filled by the method proposed in this paper was 0.95 and there were no spatial conflicts such as symbols overlapping the boundary.

(3) The computing time of the 174 forest areas of our method was 2.12 s.

The setting of the interval threshold of three-square shaped symbols had a significant influence on the distribution characteristics of the filling result. Future research will focus on exploring the statistical distribution of symbol interval parameters for processing complex polygons and large data volumes.

**Author Contributions:** Y.Y. proposed the original concept for the study. All co-authors conceived and designed the methodology. C.L., P.W., and Y.Y. were responsible for the processing and analysis of data. C.L. and P.W. drafted the manuscript. All authors read and approved the final manuscript.

**Funding:** This research was funded by the National Natural Science Foundation of China under grant number 41871375.

**Acknowledgments:** We are grateful to the anonymous reviewers whose comments helped to improve our paper. We also grateful to Xiaoli Liu for her valuable comments on this article.

**Conflicts of Interest:** No potential conflict of interest was reported by the authors.

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
