# Peer review of "Dot Symbol Auto-Filling Method for Complex Areas Considering Shape Features"

_ijgi, doi:10.3390/ijgi8030158_

Round 1

Reviewer 1 Report

The method presented in this paper for dot filling is interesting, and as far as I know new. 

One thing that should be included in the Experiment section is a discussion about the computational complexity of the method and some examples of processing times for some tests. Is this method applicable for real-time rendering? It is also good to have a figure in the Experiment section including some more polygons that have been filled with the method.

The aim of the paper should be more clearly stated.

One thing that should be discussed is whether a regular pattern is preferable for the dots. In manually made maps there are sometimes non-regular patterns for points/symbols in e.g. land use polygons. One study that mimic this manual approach is given in:

https://www.tandfonline.com/doi/abs/10.1179/000870407X241809 

A main concern is the description of the method. This description must be improved. Below is a list of items that needs to be improved.

* The description of the method in the abstract is unclear. It should be more in style of the description of the methodological section, about row 126-130

* row 14. What is meant by "areas with three types"

* row 15: What is internal and external buffer constraint

* 19: filling fullness?

* Figure 1 + row 88: The vertical blue line is not correct

* 102-105 + Figure 2: Uncleardefinition of Type1-3 triangles

* Section 3: State clearly which are the input parameters to the method

* 135: Should be stated that the original border lines a segments that goes into the constraint Delaunay triangulation.

* 144: What is meant by: topological nodes being greater than 2? This should be described by "the number of skeleton lines to the node" instead.

* 160: Here you need a description of segment extraction before you go into details.

* 164: "wide" should be "narrow"

* Figure 4: An illustration of where the triangle set A, B, C and D are is missing.

* 192: Do not get how the segments are divided into three types.

* 198: How is the origin for symbol filling defined? It is used in row 201 and 202 but I cqnnot figure out how you found the value of x0 and y0.

* 210: Here you should add an introductory paragraph to boundary constraint symbol correction. 

* What is the WJ-III map work station? Add a link/reference to explain.

* 246, 255: Avoid use the term "superiority" here and elsewhere. What you do is to validate your method and evaluate it against other methods.

* Table 1: Is the Fullness really 0. for the traditional method?

Some language comments:

row 13:"i.e." should be replaced with "denoted"

Author Response

Dear reviewer,

Thank you very much for reviewing our manuscript, and also your valuable comments. We discussed these comments carefully and revised the manuscript point by point. Please see the detailed answers in the attaced file.

Kind regards,

Yong Yin

Reviewer 2 Report

Thank you for this contribution on a new and outstanding dot symbol auto-filling method, which is of course a very specific detail for precise cartographic communication/styling. 

The paper has a very well soundness and well defined central theme. The sequence of argumentation is easy to read. 

The relevance for adding this new method is well explained. There is a handful of references given. From my point of view much more references exist. Please have a look at papers from cartographers and scientists from Suisse, England, U.S. and Australia. Within the International Cartographic Association you will find much more references on this methodology. This will enhance the relevance for this contribution. 

Author Response

Dear reviewer,

Thank you very much for reviewing our manuscript, and also your valuable comments. We discussed these comments carefully and revised the manuscript point by point. Please see the detailed answers in the attached file.

Kind regards,

Yong Yin

Round 2

Reviewer 1 Report

The paper has been improved from previous version. I have only minor comments at this stage:

Row 35: De Geer maps is interesting. But in his map each dot really represent a fixed amount of persons. That application is quite far from dot symbol filling.

Row 40: "Lars" should be "Harrie"

Row 51: Remove initial "B" in author

Row 55: de Berg, not De Berg

Row 158: You should also add the threshold values At and Wt as input values.

Row 225: length -> width (?)

Author Response

Dear reviewer,

Thank you very much for further reviewing our manuscript, and also your valuable comments. We discussed these comments carefully and revised the manuscript point by point. Please see the detailed answers in the attached file.

Kind regards,

Chengming Li

ISPRS Int. J. Geo-Inf. EISSN 2220-9964 Published by MDPI AG, Basel, Switzerland RSS E-Mail Table of Contents Alert
Back to Top